**Subglacial hydrological control on flow of an Antarctic Peninsula palaeo-ice stream**
Robert D. Larter[1*], Kelly A. Hogan[1], Claus-Dieter Hillenbrand[1], James A. Smith[1], Christine L.
Batchelor[2], Matthieu Cartigny[3], Alex J. Tate[1], James D. Kirkham[1,2], Zoë A. Roseby[4,1], Gerhard Kuhn[5],
Alastair G.C. Graham[6], and Julian A. Dowdeswell[2]
[1]British Antarctic Survey, Madingley Road, High Cross, Cambridge CB3 0ET, UK.
[2]Scott Polar Research Institute, University of Cambridge, Lensfield Road, Cambridge CB2 1ER, UK.
[3]Department of Geography, South Road, Durham University, Durham DH1 3LE, UK.
[4]Ocean and Earth Science, National Oceanography Centre, University of Southampton Waterfront
Campus, European Way, Southampton SO14 3ZH, UK.
[5]Alfred Wegener Institute, Helmholtz-Centre for Polar and Marine Research, D-27568 Bremerhaven,
Germany.
[6]College of Life and Environmental Sciences, University of Exeter, Rennes Drive, Exeter EX4 4RJ, UK
**Correspondence:** Robert D. Larter (rdla@bas.ac.uk).
**Abstract.** Basal hydrological systems play an important role in controlling the dynamic behaviour of
ice streams. Data showing their morphology and relationship to geological substrates beneath
modern ice streams are, however, sparse and difficult to collect. We present new multibeam
bathymetry data that make the Anvers-Hugo Trough west of the Antarctic Peninsula the most
completely surveyed palaeo-ice stream pathways in Antarctica. The data reveal a diverse range of
landforms, including streamlined features where there was fast flow in the palaeo-ice stream,
channels eroded by flow of subglacial water, and compelling evidence of palaeo-ice stream shear
margin locations. We interpret landforms as indicating that subglacial water availability played an
important role in facilitating ice stream flow and controlling shear margin positions. Water was likely
supplied to the ice stream bed episodically as a result of outbursts from a subglacial lake located in
the Palmer Deep basin on the inner continental shelf. These interpretations have implications for
controls on the onset of fast ice flow, the dynamic behaviour of palaeo-ice streams on the Antarctic
continental shelf, and potentially also for behaviour of modern ice streams.


## 1. Introduction

There is growing evidence that basal hydrology is a critical factor controlling the dynamic behaviour of ice streams (Bell, 2008; Christianson et al., 2014; Christoffersen et al., 2014), which account for most of the mass loss from large ice sheets. Understanding ice stream dynamics, including basal hydrology, is thus essential for quantifying ice sheet contributions to sea level change. Subglacial lakes and areas of elevated geothermal heat flux have been discovered in the onset regions of several large ice streams (Fahnestock et al., 2001; Bell et al., 2007; Stearns et al., 2008). Obtaining high resolution topographic data from modern ice stream beds that can reveal features associated with subglacial water flow is, however, logistically difficult and time consuming (e.g. Christianson et al., 2012). In contrast, modern ship-mounted sonar systems can be used to obtain such data efficiently over extensive areas of former ice stream beds on continental shelves that ice has retreated from since the Last Glacial Maximum (LGM; 23-19 k cal yr BP).

Knowledge and interpretations of submarine glacial landforms have advanced rapidly over the past two decades (Dowdeswell et al., 2016). For example, it is now recognised that elongated drumlins and mega-scale glacial lineations (MSGL) are signatures of past streaming ice flow on wet-based, mainly sedimentary beds, and that elongation generally increases with increasingly fast past flow rates (Stokes and Clark, 1999; Ó Cofaigh et al., 2002). The degree of preservation of fields of MSGL and the extent to which they are overprinted by grounding zone wedges and transverse moraines provides an indication to the rapidity of past grounding line retreat (Ó Cofaigh et al., 2008). 'Hill-hole pairs' and sediment rafts are recognised as features formed by glaciotectonic processes beneath cold, dry based ice (Evans et al., 2006; Klages et al., 2015). Erosion of meandering or anastomosing seabed channels and tunnel valleys with reversals of gradient along their length requires a hydrological pressure gradient that indicates they could only have formed beneath grounded ice (Ó Cofaigh et al., 2002; Nitsche et al., 2013). Simple hydrological pressure calculations indicate that a gentle ice surface gradient produces a pressure gradient at the bed that will drive water up an opposing bed slope nearly ten times as steep.

Modern ice sheet observations have revealed increases in ice flow rates over timescales of days to years in response to Antarctic subglacial drainage events (Stearns et al., 2008; Siegfried et al., 2016). Responses of glaciers in southwest Greenland to seasonal drainage of supraglacial meltwater to the bed, however, show that the mode of subglacial drainage is important, as a slow-down of glacier flow above a certain run-off threshold has been interpreted to correspond to a switch to more efficient, channelized drainage (Sundal et al., 2011).

Here we present extensive new multibeam bathymetry data from the Anvers-Hugo Trough (AHT)
west of the Antarctic Peninsula (Fig. 1). We interpret bedforms revealed by these data as evidence of
a basal hydrological system that influenced the flow and lateral extent of the palaeo-ice stream and
was fed by a subglacial lake in a deep basin on the inner continental shelf. We use heritage
multichannel seismic (MCS) and deep tow boomer (DTB) data to constrain the nature of substrates
beneath the LGM deposits and their potential influence on the basal hydrological system and
sediment supply.

**1.1    Glacial history and setting**

Drilling results and seismic reflection profiles indicate that the Antarctic Peninsula Ice Sheet has
advanced to its western continental shelf edge many times since the late Miocene (Larter et al.,
1997; Barker and Camerlenghi, 2002). Through repeated glaciations the ice sheet has eroded and
over-deepened the inner shelf, extended the outer shelf through progradation and delivered large
volumes of sediment to the deep ocean (Barker and Camerlenghi, 2002; Bart and Iwai, 2012;
Hernández-Molina et al., 2017). The AHT, a 140 km-long by 50 km-wide cross-shelf trough (Fig. 1),
was a recurring ice stream pathway during glacial maxima (Larter and Cunningham, 1993; Larter and
Vanneste, 1995). The most recent grounding zone advance to the shelf edge along the AHT occurred
during the LGM (Pudsey et al., 1994; Heroy and Anderson, 2005; Ó Cofaigh et al., 2014). To the
southeast of the trough, the inner shelf is incised by an erosional basin, Palmer Deep (PD), that
measures 26 km east-to-west and 10 km north-to-south at the 800 m depth contour (Domack et al.,
2006; Fig. 1). PD has a maximum depth >1400 m, yet to both north and south there are small islands
within 12 km of its axis and there is a bank directly to its west that rises to <200 metres below sea
level. Rebesco et al. (1998) and Domack et al. (2006) argued that in colder temperatures than today
and with lower sea level – e.g. at the start of Marine Isotope Stage 2 (29 ka) - ice encroached
towards PD from the nearby land areas and local ice caps formed on emergent platforms around the
present-day islands near the basin. These authors further hypothesized that continued glacial
development led to the PD basin becoming completely encircled by grounded glacial ice and to
formation of an ice shelf over it, trapping a subglacial lake. Based on multibeam bathymetry data
from the inner shelf, Domack et al. (2006) described channels crossing the deepest part of the sill
separating the western end of PD from the AHT that are 200–500 m-wide, 100–300 m deep and
exhibit reversals in their longitudinal profiles. On the basis of these characteristics and similarities to
channels in Pine Island Bay and to the Labyrinth channels in Wright Valley in the Transantarctic
Mountains that had previously been interpreted as having been eroded by subglacial water flow
(e.g. Lowe and Anderson, 2003; Nitsche et al., 2013; Lewis et al., 2006), Domack et al. (2006)
interpreted the channels as having been eroded by outflow from a subglacial lake in PD. More
recently, geochemical analysis of pore waters from sediments in one of the basins within the
channel network in Pine Island Bay confirmed that it had been a sub-glacial lake (Kuhn et al., 2017).

**2.   Methods**

**2.1      Multibeam bathymetry and acoustic sub-bottom profile dat**a

Extensive new data were collected on RRS *James Clark Ross* cruise JR284 in January 2014 using a
1°x1° Kongsberg EM122 system with 432 beams and a transmission frequency in the range 11.25–
12.75 kHz. Beam raypaths and sea bed depths were calculated in near real time using sound velocity
profiles derived from conductivity temperature-depth and expendable bathythermograph casts
made during the cruise. Processing consisted of rejecting outlying values, replacing the sound
velocity profile applied during acquisition with a more relevant one for some data, and gridding to
isometric 30 m cells using a Gaussian weighted mean filter algorithm in MB-System software (Caress
and Chayes, 1996; Caress et al., 2018). Pre-existing multibeam data, mostly collected on RVIB
*Nathaniel B. Palmer* and previous cruises of RRS *James Clark Ross* (Anderson, 2005; Domack, 2005;
Lavoie et al., 2015), and data along a few tracks collected more recently on HMS *Protector* using a
Kongsberg EM710 system (70–100 kHz), were included in the grid. Acoustic sub-bottom echo
sounding profiles were collected along all JR284 survey lines with a Kongsberg TOPAS PS018
parametric system using a 15 ms chirp transmission pulse with secondary frequencies ranging 1.3–5
kHz. Vessel motion and GPS navigation data on cruise JR284 were collected using a Seatex Seapath
system.

**2.2      Heritage seismic reflection data**

MCS data used were collected in the 1980s on RRS *Discovery* cruises D154 and D172 (Larter and
Cunningham, 1993, Larter et al., 1997). On D154, Line AMG845-03 was collected with a 2400 m-long
hydrophone streamer, whereas on D172 the streamer used to collect data on Line BAS878-11 was
800 m in length. In both cases the streamer was towed at 8–10 m depth and data were recorded
from 50 m-long groups with a sampling interval of 4 ms. The seismic source consisted of four airguns
with total volumes of 8.5 l on D154 and 15.8 l on D172, respectively, and data were processed to
common mid-point stack using standard procedures. Very high resolution seismic data were

collected using a Huntec DTB, towed within 100 m of the sea bed, on RRS *James Clark Ross* cruise JR01 in 1992 (Larter and Vanneste, 1995; Vanneste and Larter, 1995). This system transmitted a broadband pulse with frequencies 0.8–10 kHz and a cycle time of 0.9 s. Data were recorded with a 100 μs sampling interval from a 1 m-long hydrophone trailed behind the towed vehicle. The DTB system is capable of resolving sedimentary layers <1 m in thickness and also achieves very high spatial resolution due to the proximity of the source and receiver to the sea bed.

**3.  Results**

**3.1      Description and interpretation of landforms and seismic/acoustic profiles**.

Integration of the new multibeam bathymetry data with pre-existing data provides nearly continuous coverage of the AHT from PD to beyond the continental shelf edge, with the new data spanning the full width of the trough on the middle shelf (Fig. 1). They also include coverage of the confluence with a tributary trough that joins the AHT from the east on the middle shelf. We will refer to this tributary by the informal name 'Perrier Trough', as it originates offshore from Perrier Bay, Anvers Island. The data reveal extensive areas of MSGL and drumlins, which are characteristic of ice stream beds and show the pattern of palaeo-ice flow (Stokes and Clark, 1999; King et al., 2009; Graham et al., 2009). Fields of drumlins, with elongation ratios between 2.5 and 6:1, occur over a broad transition zone between the rugged inner shelf and smoother mid-shelf part of the AHT, and where the AHT crosses a structural high that separates middle and outer shelf areas (Larter et al., 1997). MSGL in the mid-shelf part of the AHT have elongation ratios between 12 and 17:1, whereas some on the outer shelf have elongation ratios up to 80:1. The data also confirm the occurrence of several grounding zone wedges (Fig. 1), some of which had been identified previously, indicating positions where the grounding zone paused during retreat from its LGM position (Larter and Vanneste, 1995; Vanneste and Larter, 1995; Batchelor and Dowdeswell, 2015). Here, however, we focus on two specific areas in which the landforms observed have a bearing on the role of subglacial hydrology in facilitating and controlling ice stream flow.

**3.2      Southern Anvers-Hugo Trough**

In the southern part of the AHT there is a marked along-trough change in landforms across a line where a MCS profile (AMG845-03) shows that a mid-shelf sedimentary basin pinches out (Figs 2a and 3), with acoustic basement cropping out to the southeast (Larter et al., 1997). The acoustic

basement most likely represents Palaeozoic-Mesozoic igneous and metasedimentary rocks similar to those that crop out on the nearby islands (Storey and Garrett, 1985; Leat et al., 1995). The sedimentary strata in the basin are of unknown age, but it has previously been inferred that the youngest layers could be no younger than middle Miocene and the oldest layers may be early Tertiary or Late Cretaceous in age (Larter et al., 1997). Crescentic scours around the 'upstream' ends of bathymetric highs and fields of anastomosing channels are observed in the area where acoustic basement crops out (Fig. 2a). Among the anastomosing channels, the largest are up to 30 m deep and 250 m wide, although many are smaller (Fig. 2b).

In the axis of the AHT directly north of this zone, incised into the edge of the sedimentary basin, the new data reveal a set of northward shoaling and narrowing valleys spaced 2–3 km apart (Fig. 2a). Individual valleys are up to 1100 m wide and 60 m deep at their southern ends (Fig. 2c), but become narrower and shallower northwards (Fig. 2c,d ), ultimately petering out over a distance of <5 km. The slope along the steepest part of the channel axes is ~2° (Fig. 2d). In detail, the southern part of each valley exhibits a v-shaped deeper section incised into a u-shaped upper section, the v-shaped sections being up to 350 m wide and 45 m deep (Fig. 2c). MSGL start directly north of these valleys (Figs 2a and 4) and cover most of the sea bed in the trough between this point and the continental shelf edge.

Two acoustic sub-bottom profiles that run transverse to the trough about 8 and 10 km north of the northern tips of the valleys show that the MSGL were formed in the surface of an acoustically transparent layer that is draped by a 4 m-thick layer of younger sediments (Fig. 4). Where such acoustically transparent layers have been cored elsewhere on the West Antarctic shelf they have been shown to consist of relatively low shear strength, deformed till, dubbed 'soft till', and the high amplitude reflector below this layer has been shown to correlate with the top of a higher shear strength 'stiff till' (e.g. Ó Cofaigh et al., 2005a; Reinardy et al., 2011). A homogenous, terrigenous diamicton with moderate shear strength, which was recovered in marine sediment cores from AHT, documents the presence of the soft till there (Pudsey et al., 1994; Heroy and Anderson, 2005). Each of the sub-bottom profiles shows three depressions in this high amplitude reflector that are 8–10 m deep, 400–800 m wide and contain an acoustically transparent fill, above which a reflector is usually observed separating the fill from the overlying soft till layer (Fig. 4).

We interpret the anastomosing channels and crescentic scours incised into the hard substrate on the inner shelf as having been eroded by subglacial water flow at times when grounded ice extended further offshore, as previous authors have interpreted similar features elsewhere (Dreimanis, 1993; Lowe and Anderson, 2003; Nitsche et al., 2013; Lewis et al., 2006; Graham and Hogan, 2016). This origin is also consistent with the interpretation that a subglacial lake was present in PD during the

last glacial period, and channels incising the sill at its western end were eroded by outflow from the
lake (Domack et al., 2006). Considering the scale of such features here, as well as in other areas
around Antarctica, and the nature of the material they are eroded into, they probably developed
progressively through multiple glacial cycles (Smith et al., 2009; Nitsche et al., 2013). The scale of the
features also implies that water flow rates fast enough to drive erosion can only have been achieved
through subglacial storage of water and episodic discharge (rather than continuous flow), even if it is
assumed that the upper parts of scours and channels were filled with ice (Nitsche et al., 2013;
Kirkham et al., 2019). This is, once again, consistent with the interpretation of a subglacial lake in PD,
and also with a semi-quantitative model that implies outbursts from trapped subglacial lakes in such
settings and of these approximate dimensions are likely to occur with repeat periods of the order of
a few centuries (Alley et al., 2006).
A conservative estimate for the volume of the PD subglacial lake stated by Domack et al. (2006)
was 20 km$^3$. However, the full volume of the basin deeper than the sill depth of ~500 m is about 110
km$^3$. If this was filled the lake would have been nearly two orders of magnitude greater in volume
than subglacial Lake Ellsworth (1.37 km$^3$, Woodward et al., 2010), but still nearly two orders of
magnitude smaller than Lake Vostock (5400 ±1600 km$^3$, Studinger et al., 2004). The length and width
of PD are similar to Lake Engelhardt, the largest of a number of connected shallow lakes beneath the
lower Mercer and Whillans ice streams, from which remote sensing data show ~2 km$^3$ of water
drained between October 2003 and June 2006 (Fricker et al., 2007). However, water depth in these
lakes likely rarely exceeds 10 metres (Christianson et al., 2012), so their volumes are small compared
to the potential size of lakes in deep basins such as PD.
Considering that the northward-shoaling valleys are located in the axis of the AHT directly north
of an area containing anastomosing channels and crescentic scours, we interpret them as also having
been eroded by subglacial water flow. We interpret the upper u-shaped sections of the valleys (Fig.
2c) as having been widened by glacial erosion, implying that the valleys have been overridden by ice
since they were first carved. This is consistent with the suggestion that many features eroded into
bedrock on Antarctic continental shelves developed through multiple glaciations (Graham et al.,
2009). Even at times when there was active water flow, ice may also have filled a large part of the v-
shaped lower sections. Palaeo-ice flow paths indicated by streamlined bedforms show that the area
in the axis of the AHT where the valleys occur lies directly down flow from the sill at the western end
of PD. MSGL directly north of the valleys indicate that there was fast ice flow in this part of the
trough, likely facilitated by the soft till layer that is seen as an acoustically transparent layer in sub-
bottom profiles (cf. Alley et al., 1986; Ó Cofaigh et al., 2005a; Reinardy et al., 2011). The coincidence
of the onset of MSGL with northward disappearance of the valleys suggests that water supplied
through them was important in lubricating and dilating the till, thus reducing its shear strength and
making it more prone to deform under stress. Thus we infer that the northward terminations of the
valleys were associated with a transition from channelized to distributed water flow at the ice bed.
The shallow, filled depressions observed in the sub-bottom profiles (Fig. 4) suggest that the valleys
once continued further to the north, before their distal reaches were filled and the overlying soft till
in which MSGL are formed accumulated beneath fast-flowing ice. The sequence of units observed in
the profiles could have resulted from upstream migration of the onset of sediment-lubricated
streaming flow during the last glaciation.

**3.3      Confluence of Anvers-Hugo and Perrier troughs**

In the area of the confluence between AHT and Perrier Trough, an area of east-west aligned MSGL
terminates abruptly along a line parallel to their trend on the southern flank of the Perrier Trough
(Fig. 5a). The eastern limit of the area covered by MSGL in AHT is more irregular, but lies within a
band no more than 3 km wide on the eastern flank of the trough. Streamlined bedforms are absent
from the area between the two troughs, but several steep-sided bathymetric basins up to 1500 m
wide and 40 m deep are observed (Fig. 5a-c). The central parts of most basins are flat or gently-
dipping so that cross-sections exhibit box-shaped profiles (Fig. 5b,c). A 500 m-wide and 8 m-high
mound occurs on the north-western side of one of the largest basins (Fig. 5c). A few of the basins
span the boundary of MSGL on the southern flank of Perrier Trough. Furthermore, about 3 km north
of the boundary of MSGL and to the east of the other basins, linear features connect a small basin
~300 m in diameter with a similarly-sized mound 1.6 km to its WNW (Fig. 5a).
We interpret the well-defined southern lateral boundary of MSGL in Perrier Trough as indicating
the marginal shear zone position when the palaeo-ice stream reached its maximum width during the
LGM. Similarly, the lateral limit of MSGL on the eastern flank of the AHT lies within a band that is no
more than 3 km wide, indicating the approximate position of the shear zone at the margin of the
palaeo-ice stream occupying this trough when it was at its widest. Such clear expressions of ice
stream shear margin positions in bed morphology have proved elusive beneath the modern ice
sheet, partly because the chaotic structure of shear zones causes scattering of ice-penetrating radar
signals that is observed as 'clutter' in survey data (Shabtaie and Bentley, 1987). The steep-sided
basins in the area between the two troughs are similar to the holes of hill-hole pairs and scars
resulting from detachment of sediment rafts observed on the northeastern part of the Amundsen
Sea continental shelf (Klages et al., 2013, 2015, 2016). The small mound and basin in the Perrier
Trough 3 km north of the boundary of MSGL appear to be a clear example of a hill-hole pair. Such

features are generally regarded as characteristic of erosion and deformation beneath dry-based ice cover (e.g. Ottesen et al., 2005; Evans et al., 2006). The mound to the northwest of one of the largest basins may represent a corresponding hill (Fig. 5a, c), although as its cross-sectional area is smaller than that of the adjacent hole it cannot contain all of the excavated sediment. The absence of mounds near to some of the other basins may be explained by their close proximity to the palaeo-ice stream confluence, as the excavated material would only need to have been transported a short distance into the path of one of the ice streams to be entrained by faster flow. Evidence of hill-hole pairs having been overridden and eroded by ice after their formation has previously been reported from the Norwegian continental shelf, where some hills are observed to have streamlined tails (Rise et al., 2016). The pristine form of the hill-hole pair within the Perrier Trough indicates that it must have formed after ice stream flow had stagnated.

A DTB profile runs across the bathymetric basin that lies closest to the confluence of the two troughs and, beneath thin, patchily distributed younger sediments, it shows an acoustically transparent layer up to 25 ms (~20 m) thick (Figs 5a, 6). This layer has a minimum thickness of <3 m beneath the south-eastern edge of the basin floor, and it thickens progressively to the north-west across the basin. In places short segments of truncated, dipping reflectors can be seen beneath the strong reflector at the base of the acoustically transparent layer (Fig. 6). We interpret the acoustically transparent layer as consisting of Quaternary diamictons overlying an unconformity above mid-shelf basin sedimentary strata that are represented by the truncated, dipping reflectors. The reduced thickness of the acoustically transparent layer across the basin suggests that the 'hole' was formed by erosion of Quaternary diamictons and that the older sedimentary strata were unaffected during its formation. As mentioned above, such holes are generally regarded as characteristic of erosion beneath dry-based ice cover, and therefore the restriction of erosion to the Quaternary diamictons is consistent with the shear stress threshold for brittle failure in them likely being significantly lower than in the underlying consolidated strata (Evans et al., 2006). At the sea bed near the foot of the steep south-east flank of the basin, a unit that is 180 m across and 7 ms (~5.5 m) thick containing south-east dipping reflectors is observed. We interpret this unit as a rotated slide block that has originated from the flank of the 'hole' after its excavation. The most prominent reflector within the block is most likely to be from the boundary between diamicton and postglacial sediment as observed to the NW of the basin (Fig, 6), in which case the displacement of the block did not occur until after a layer of postglacial sediments several metres in thickness had accumulated.

A MCS profile that runs obliquely across Perrier Trough and continues over the inter-trough area shows mid-shelf basin sedimentary strata underlying the sea bed along the entire line (Figs 5a, 7). As

noted above, the oldest strata in this basin may be as old as Late Cretaceous, but most of the basin fill is likely of Tertiary age (Larter et al., 1997). The southern lateral limit of MSGL in Perrier Trough lies within 1 km of the position where a unit of younger strata with a distinct seismic facies character (labelled 'later mid-shelf basin' in Fig. 7) pinches out, but is not coincident with this boundary.

**4. Discussion and Conclusions**

The features described here suggest that subglacial water, likely supplied episodically from a subglacial lake in PD, played an important role in facilitating ice stream flow in the AHT during the last and probably several late Quaternary glacial periods, and likely modulated the flow velocity.

In the palaeo-ice stream confluence area (Fig. 5a) the close juxtaposition of MSGL, which are characteristic of wet-based, fast ice flow (Stokes and Clark, 1999; King et al., 2009; Graham et al., 2009), with excavated basins ('holes') that are characteristic of slow, dry-based ice flow (Ottesen et al., 2005; Evans et al., 2006), suggests that water availability was an important control on the lateral extent of the palaeo-ice streams. This interpretation is supported by a MCS profile that shows the palaeo-ice stream shear margin position on the south side of Perrier Trough does not coincide with a major geological boundary (Fig. 7). The MCS profile shows that the Perrier Trough palaeo-ice stream and the inter-stream area are both underlain by dipping sedimentary strata of likely Tertiary age. The position of the shear margin does, however, lie within 1 km of a second-order boundary between strata of different ages and with different seismic facies that may have had some influence on its position (Fig. 7). Unfortunately, available seismic profiles are too widely spaced to assess how closely the palaeo-shear margin follows this boundary.

The observation that a few of the 'holes', and one clear hill-hole pair, span the boundary of, and cross-cut, MSGL on the southern flank of Perrier Trough suggests inward ice stream shear margin migration during glacial recession. Lateral migration of shear margins was inferred in the area near the grounding on the modern Thwaites Glacier between 1996–2000 (Rignot et al., 2002), although a later study concluded that there had been no significant migration of the eastern shear margin of the glacier during the most recent two decades (MacGregor et al., 2013). On different parts of the northern shear margin of Whillans Ice Stream, migration rates of up to 280 ma$^{-1}$ outwards and up to 170 ma$^{-1}$ inwards were measured over the decade to 1997 (Stearns et al., 2005). Ice-penetrating radar data across the northern margin of Kamb Ice Steam suggest abrupt inward migration of the margin ~200 years before complete stagnation flow, attributed to reduced lubrication (Catania et al., 2006). Reduction in basal water supply could occur due to depletion of upstream reservoirs (cf. Christoffersen et al., 2014), due to surface slope reduction leading to subglacial flow

change/reversal,  or due to ice thinning or decreasing flow rate and consequent reduction in
pressure melting or strain heating, respectively. Alternatively, in the case of palaeo-ice streams on
continental shelves, reduction in basal water supply could have resulted from total evacuation of
subglacial lakes trapped during ice advance at the LGM. The recovery of tills older than 13 cal. ka BP
at the bases of sediment cores from PD indicates that the lake there was eventually evacuated
(Barker & Camerlenghi 2002; Domack et al. 2001, 2006). However, water discharged from PD would
not have supplied the bed of the palaeo-ice stream in Perrier Trough, so lake evacuation could only
explain decreased water supply there if subglacial lakes were trapped in other deep basins upstream
of the area studied.

The onset of MSGL in AHT coincides with the downstream termination of the northward shoaling

valleys. This spatial coincidence suggests that water delivered through the channels played a role in
promoting streaming ice flow northward of this point. The upstream dip of 2° along the steepest
part of the channels (Fig. 2d) implies the minimum ice surface gradient required to produce a basal
hydrological pressure gradient that would drive water northward along the valleys was only 0.2°,
which is within the range of surface gradients on many modern ice streams (Horgan and
Anandakrishnan, 2006). Water that flowed along the valleys may have been either incorporated into
the till layer that the MSGL formed in, thereby dilating it and facilitating shear deformation, or
dissipated into a thin film that spread along the ice-sediment interface (cf. Ó Cofaigh et al., 2005a).
The sudden appearance of MSGL at this point also requires a source for the till itself, and the most
obvious candidate is the underlying strata at the edge of the sedimentary basin (Fig. 3). Although
these strata have never been sampled they are the most likely source of reworked Cretaceous
radiolarians found in diamictons recovered by drilling on the outer shelf (Shipboard Scientific Party,
1999).Onset of MSGL where ice flowed onto a bed consisting of older sedimentary strata has also
been reported in other locations (e.g. Wellner et al., 2001, 2006, Graham et al., 2009). Erosion of
material from underlying sedimentary strata by, or in the presence of, subglacial water flow presents
a potential mechanism for generating a dilated basal till layer of the kind that has been shown to be
present beneath some modern ice streams (e.g. Alley et al., 1986; Smith, 1997).

Subglacial lakes in deep inshore basins such as PD are likely to form during an ice sheet growth

phase (Domack et al., 2006; Alley et al., 2006). In this case, episodes of water expulsion from PD may
have accelerated ice flow and thus contributed to rapid advance of grounded ice with a low surface
gradient across the shelf. Acceleration of ice flow by about 10% over 14 months was observed on
Byrd Glacier following a drainage event from subglacial lakes 200 km upstream of the grounding line
(Stearns et al., 2008), and subglacial meltwater drainage flux and routing is also known to influence
flow rates of glaciers in Greenland (Sundal et al., 2011). The axis of AHT shallows steadily from >600
m where MSGL start on the middle shelf to <440 m at the shelf edge (Vanneste and Larter, 1995),
and grounding lines are potentially unstable on such upstream-deepening beds (Weertman, 1974;
Schoof, 2007; Katz and Worster, 2010). Therefore, if episodic subglacial water outbursts caused
'surging' ice stream behaviour, as envisaged by Alley et al. (2006), they may also have resulted in
fluctuations in grounding line positions.

Our interpretations are consistent with the hypothesis that subglacial lakes or areas of elevated

geothermal heat flux play a critical role in the onset of many large ice streams (Bell, 2008). There are
other deep, steep-sided inner shelf basins where subglacial lakes could have been trapped during
glacial advance in the catchment areas of several other well-documented Antarctic palaeo-ice
streams. For example, there are several basins >900 m deep in Marguerite Bay, which is part of the
catchment of the Marguerite Trough palaeo-ice stream (Livingstone et al., 2013; Arndt et al., 2013),
and there is a >1100 m-deep basin in Eltanin Bay at the head of the Belgica Trough palaeo-ice stream
(Ó Cofaigh et al., 2005b; Graham et al., 2011; Arndt et al., 2013). In the Amundsen Sea Embayment,
there are >1500 m-deep inner shelf basins in the catchments of both the Pine Island-Thwaites and
Dotson-Getz palaeo-ice streams (Larter et al., 2009; Graham et al., 2009, 2016; Nitsche et al., 2013;
Arndt et al., 2013; Witus et al., 2014), and it has been shown that at least one of those in the former
area did indeed host a sub-glacial lake (Kuhn et al., 2017). Hence subglacial lakes at the onset of
many continental shelf palaeo-ice streams may have facilitated their advance across the shelf during
late Quaternary glacial periods. Furthermore, if the lakes persisted when the ice streams had
advanced to the outer shelf, outbursts from them could have caused surge-like behaviour leading to
fluctuations in grounding line positions on typical inward-deepening Antarctic continental shelf
areas. Such behaviour of marine-based palaeo-ice streams on timescales of the order of centuries, as
suggested by the simple model proposed by Alley et al. (2006), could explain the observation of
cross-cutting MSGL in several outer continental shelf areas (e.g. Ó Cofaigh et al., 2005a, 2005b;
Mosola and Anderson, 2006). The preservation of MSGL from successive flow phases precludes
erosion or deposition of more than a few metres of sediment between them, which is easier to
envisage if the time separation between the flow phases is relatively small. The potential for such
subglacial lake outbursts to recur at decadal to centennial intervals and to cause significant ice
dynamic fluctuations means that there is a need to better understand the processes involved in
order to better forecast the future behaviour of modern ice streams and the contribution they will
make to sea-level change.

*Data availability.* The multibeam bathymetry grid is available online from the UK Polar Data Centre
(https://doi.org/10.5285/70905d9c-6dc0-421b-b20b-1e2dff97e802). The raw multibeam data and
acoustic sub-bottom profiler data collected on cruise JR284, as well as the heritage seismic data, are
available on request from the UK Polar Data Centre. Stacked MCS data from line AMG845-03 are
available from the Scientific Committee on Antarctic Research Seismic Data Library System
(http://sdls.ogs.trieste.it/).

*Author contributions.* RDL conceived the idea for the study and together with C-DH, JAS, JAD, KAH
and AGCG developed the research plan for cruise JR284. RDL together with KAH, C-DH and JAS wrote
the manuscript. RDL, KAH, AJT, CLB, C-DH, JAS and MC collected the multibeam bathymetry and
acoustic sub-bottom profile data on cruise JR284. JDK, ZAR, GK, AGCG and JAD participated in
discussions with the aforementioned authors on interpretation of the data and their implications for
subglacial hydrological systems. All authors commented on the manuscript and provided input to its
final version.

*Competing interests.* The authors declare that they have no conflict of interest.

*Acknowledgements.* We thank the captain, officers and crew on RRS *James Clark Ross* cruise JR284,
and Elanor Gowland and Ove Meisel who assisted with data collection. We also thank the captain,
hydrographers, officers and crew who sailed on HMS *Protector* during the 2015-16 and 2016-17
Antarctic seasons, from which additional data were incorporated. Heritage data collected on RVIB
*Nathaniel B. Palmer* were obtained from the Marine Geoscience Data System (www.marine-
geo.org). The MCS data on line BAS878-11 were processed by Alex Cunningham, and Christian dos
Santos Ferreira provided valuable help and advice in using MB-System to suppress some artefacts.
This study is part of the Polar Science for Planet Earth Programme of the British Antarctic Survey.
Participation of KAH, CLB and MC on cruise JR284 was funded by Natural Environment Research
Council Collaborative Gearing Scheme awards. We thank Huw Horgan and an anonymous reviewer
for helpful reviews that improved the paper.

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

 **Figures**


Fig. 1. Multibeam bathymetry data over Anvers-Hugo Trough, Perrier Trough and Palmer Deep. Grid-
cell size 30 m, displayed with shaded relief illumination from northeast. Regional bathymetry
contours from IBCSO v1.0 (Arndt et al., 2013). Red dashed lines mark interpreted past grounding
zone positions, with earliest ones labelled GZ1–GZ3. Most upstream grounding zone in Perrier
Trough is labelled GZ3P, as it did not necessarily form synchronously with GZ3. Only discontinuous
segments of later grounding zone positions are identified in Anvers-Hugo Trough. Black boxes show
locations of Figs 2a and 5a. Red box on inset shows location of main figure.

Fig. 2. **a** Detail of multibeam bathymetry over the boundary of the mid-shelf basin in the southern
part of Anvers-Hugo Trough. Grid-cell size 30 m, displayed with shaded relief illumination from 075°.
Dashed red lines mark interpreted former grounding zone positions. Purple line marks location of
MCS profile in Fig. 3, with small dots at intervals of 10 shots and larger dots and annotations at 100-
shot intervals. White lines mark locations of topographic profiles in **b-d** and solid red lines mark
locations of acoustic sub-bottom profiles in Fig. 4. Semi-transparent pink-filled areas crossing the
sub-bottom profiles mark the positions of the buried channels observed in the profiles and
interpolated between them. Yellow dotted line outlines approximate extent of area of anastomosing
channels. MSGL, mega scale glacial lineations. **b** Profile across anastomosing channels. **c** Cross
sections of northward shoaling valleys. **d** Profile along axis of one northward shoaling valley.
Locations of profiles shown in **a**.

Fig. 3. Part of MCS Line AMG845-03 and interpreted line drawing, showing sedimentary basin pinch
out at ~SP530. Dotted grey lines labelled B on the line drawing mark prominent bubble pulse
reverberations following the sea-floor reflection. Location of profile, including shot point positions,
shown in Fig. 2a.

Fig, 4. Acoustic sub-bottom (TOPAS) profiles from the southern part of Anvers-Hugo Trough showing
buried, filled channels overlain by acoustically-transparent 'soft till' layer with MSGL on its surface,
which is in turn overlain by a thin drape of postglacial sediments. Locations of profiles shown in Fig.
2a.

Fig. 5. **a** Detail of multibeam bathymetry over the confluence between Anvers-Hugo Trough and
Perrier Trough, which joins it from the east. Grid-cell size 30 m, displayed with shaded relief
illumination from northeast. Black lines marks locations of topographic profiles in **b-c**. Red line marks
location of DTB profile in Fig. 6. Purple line marks location of MCS profile in Fig. 7, with small dots at
intervals of 10 shots and larger dots and annotations at 100-shot intervals. Dark blue dashed lines
outline selected box-shaped bathymetric basins. AHT, Anvers-Hugo Trough; MSGL, mega scale glacial
lineations; GZ2, GZ3P, interpreted former grounding zone positions marked by dashed red lines.
Orange dotted line marks lateral limit of MSGL, interpreted as position of palaeo-ice stream shear
margins. **b** Cross sections of box-shaped basins running approximately SW-NE (i.e. approximately
transverse to the inferred palaeo-ice flow direction). **c** Cross-section of one of the larger basins in an
approximately NW-SE direction. Locations of profiles shown in **a**.

Fig. 6. Part of DTB Line 4 across the step-sided basin that lies closest to the trough confluence. The
profile shows an acoustically-transparent layer of variably thickness, interpreted as Quaternary
diamicton, overlying a strong reflector, interpreted as an unconformity above older sedimentary
strata. Short segments of truncated, dipping reflectors, marked by upward pointing small arrows,
can be seen beneath the strong reflector at the base of the acoustically-transparent layer. Two-way
travel times have been corrected for the tow depth of the boomer so that they represent
approximate times from the sea surface. Location of profile shown in Fig. 5a.

Fig. 7. Part of MCS Line BAS878-11 and interpreted line drawing, showing the entire area of the
confluence between Anvers-Hugo Trough and a tributary trough is underlain by sedimentary strata,
and that the lateral limit of MSGL in the tributary trough lies within 1 km of the position where a unit
of younger strata with a distinct seismic facies character pinches out. An f-k demultiple process was
used to suppress the sea-floor multiple reflection on this line. Dotted grey lines labelled B on the line
drawing mark prominent bubble pulse reverberations following the sea-floor reflection. Location of
profile shown in Fig. 5a.

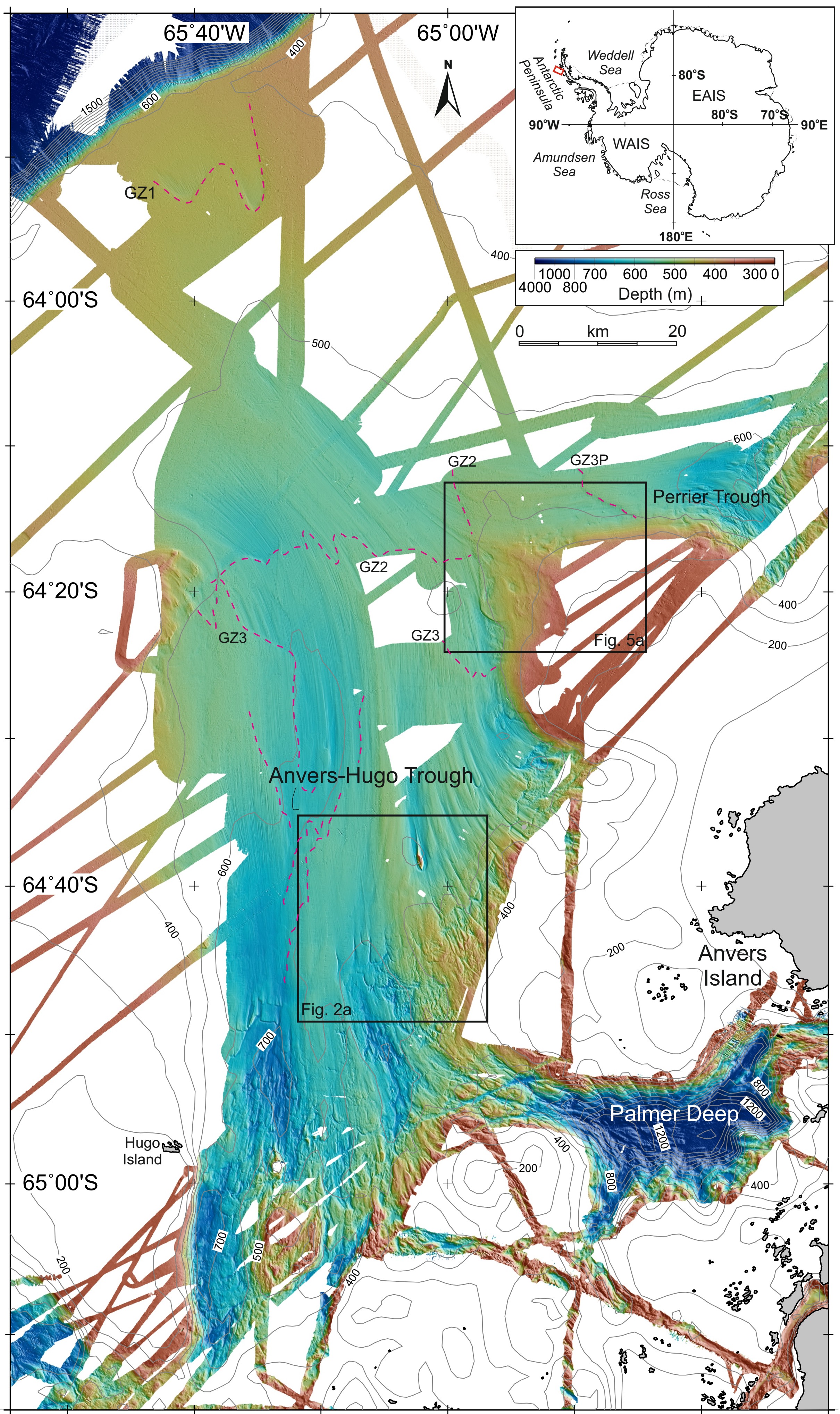

65°40'W    65°00'W

N

GZ1

64°00'S

500

400

GZ2    GZ3P
Perrier Trough

GZ2

64°20'S

GZ3    GZ3

Fig. 5a

400

200

Anvers-Hugo Trough

600

64°40'S

400

Fig. 2a

Anvers
Island

200

700

Hugo
Island

Palmer Deep

65°00'S

1200

800

400

700
500

200

400

Weddell
Sea

Antarctic
Peninsula

80°S

EAIS

90°W    80°S    70°S    90°E

WAIS

Amundsen
Sea

Ross
Sea

180°E

1000  700  600  500  400  0
4000  800    Depth (m)

0    km    20

Fig. 1

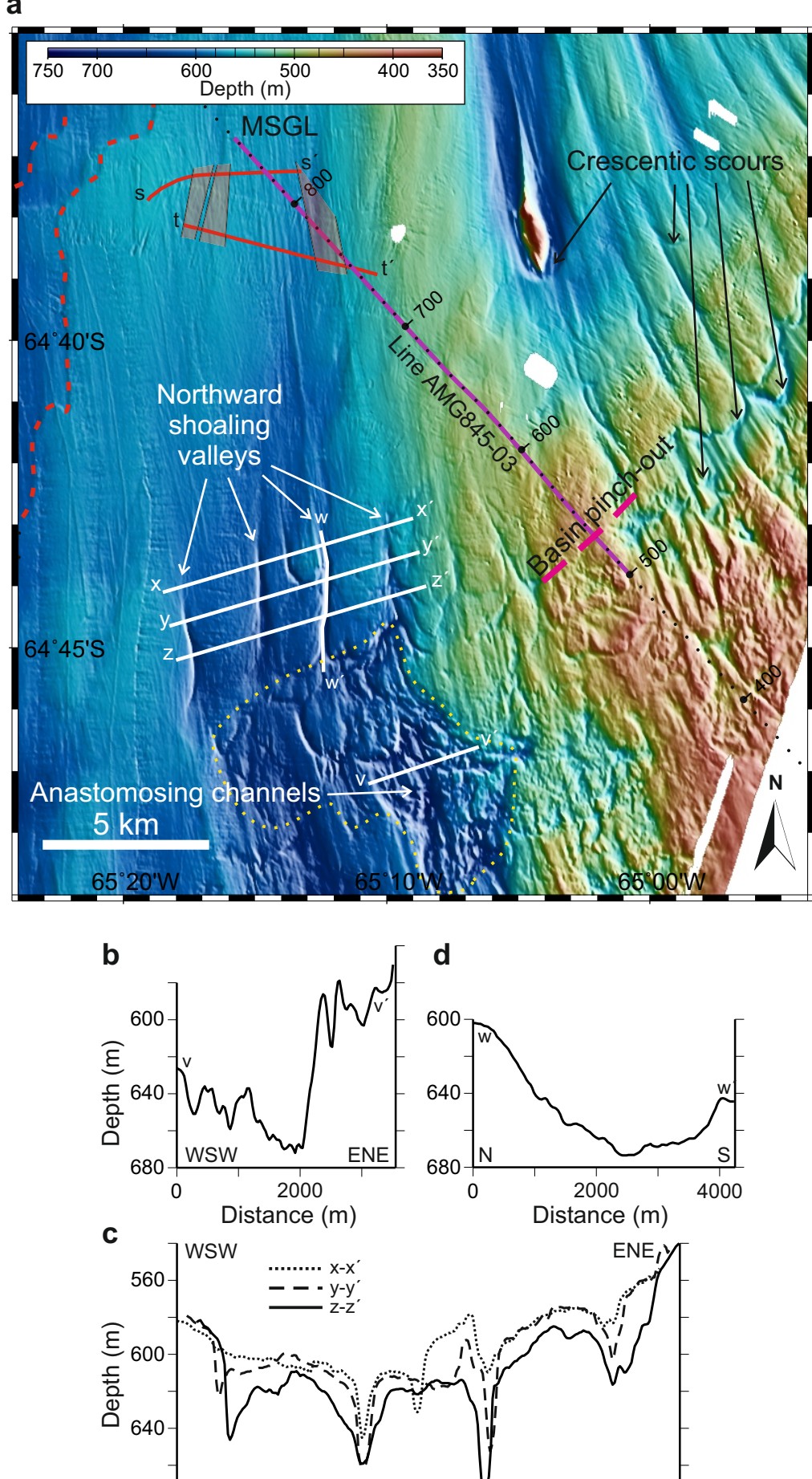

Fig. 2

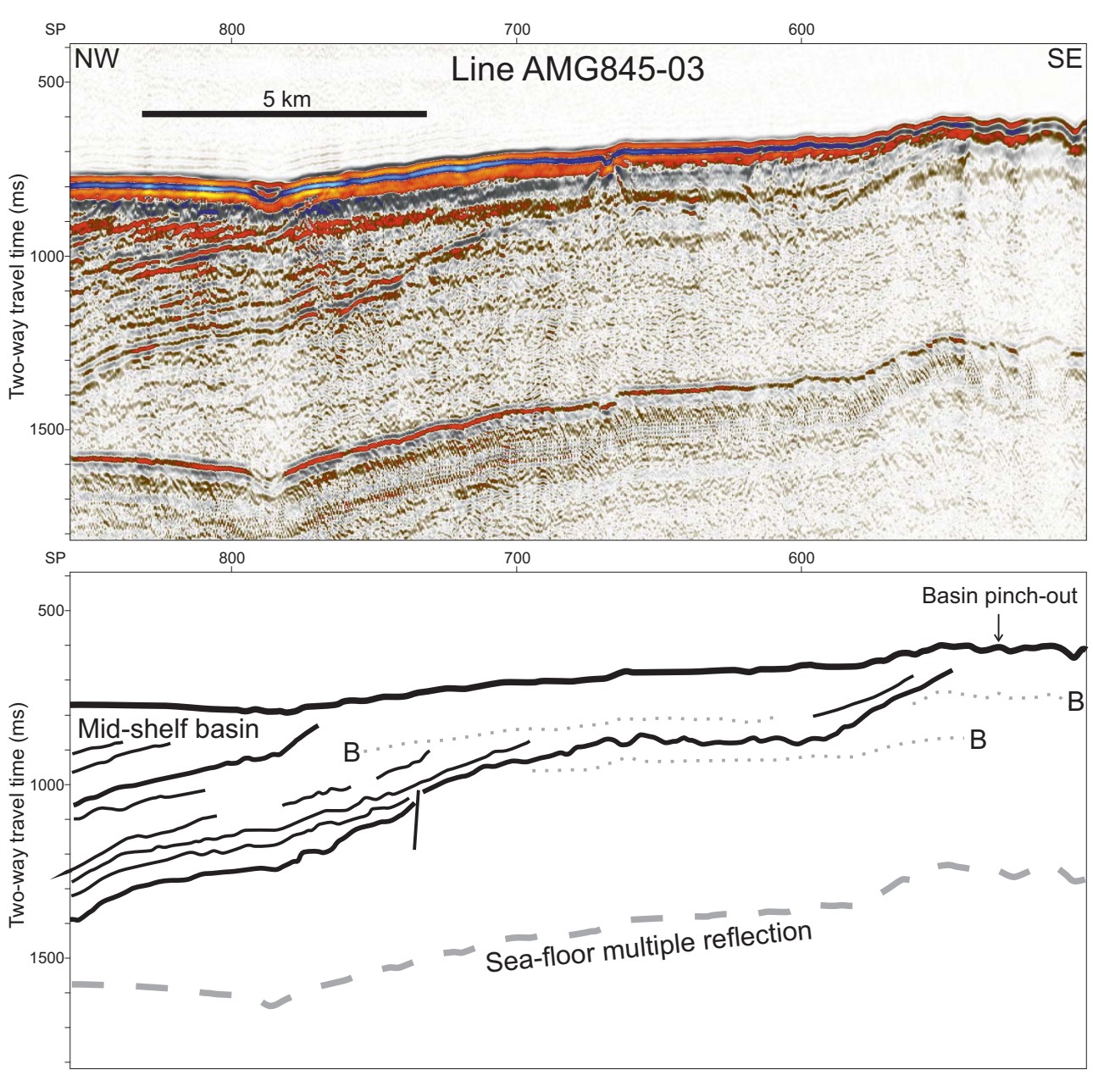

Fig. 3

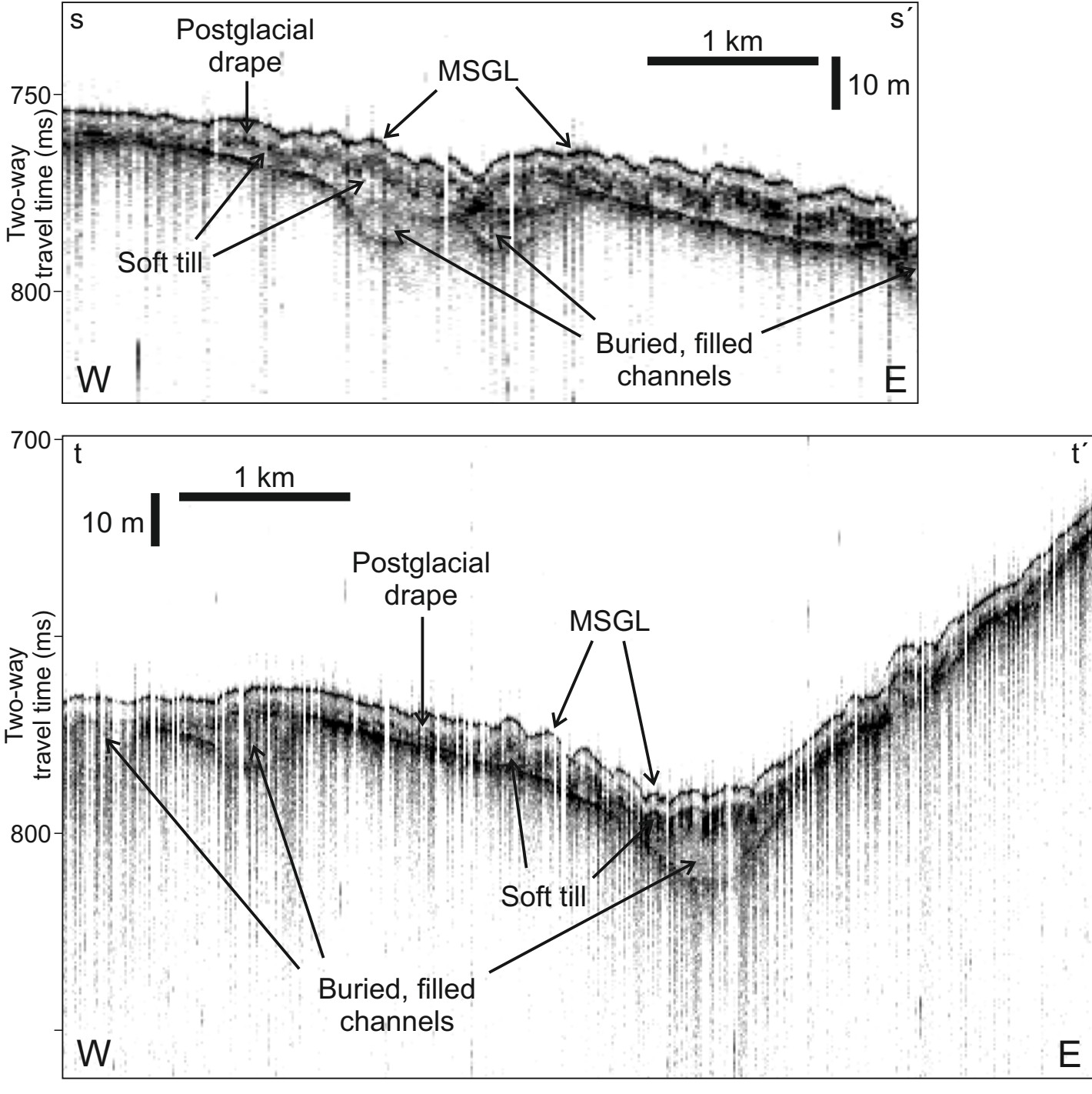

Fig. 4

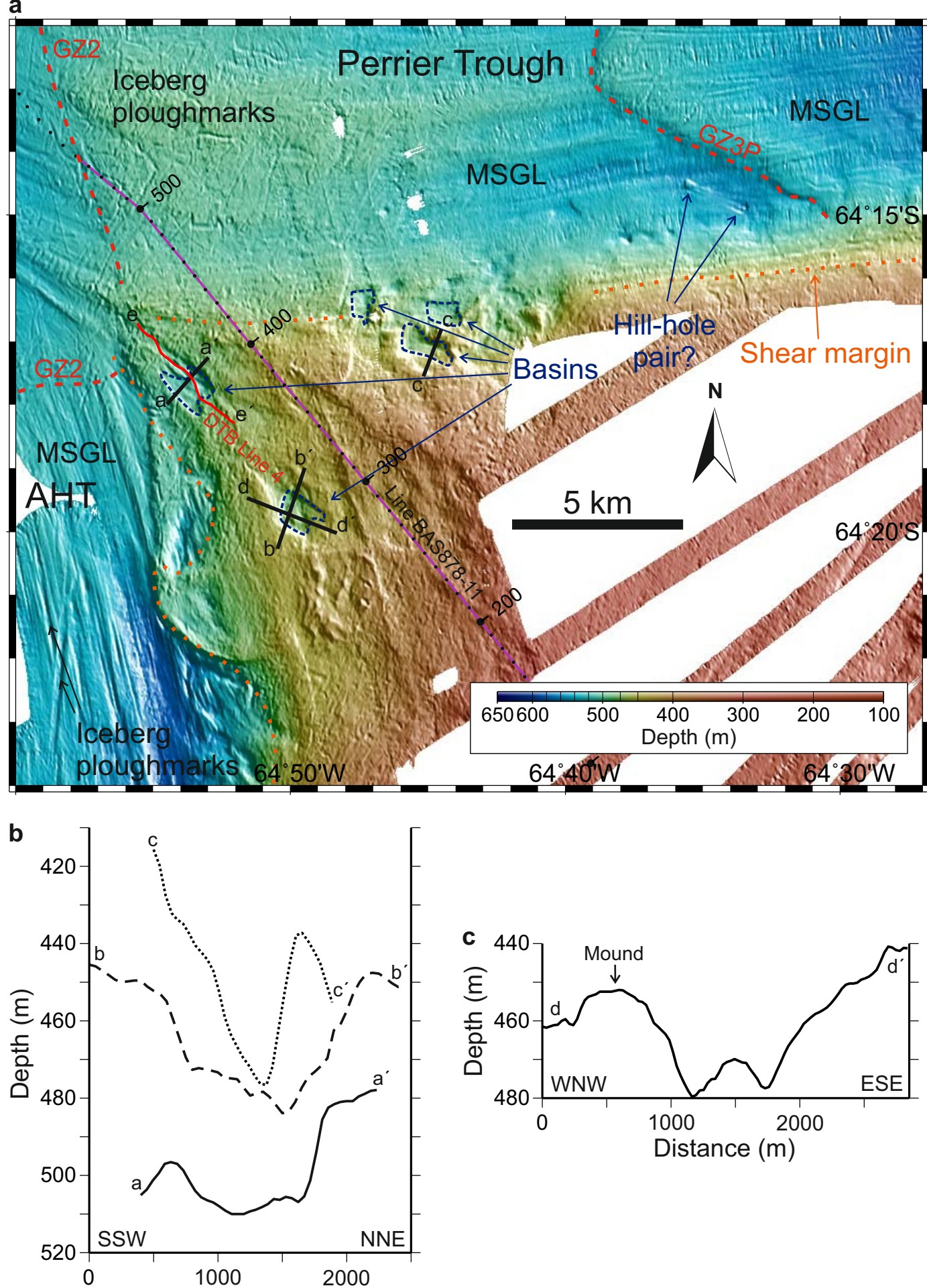

Fig. 5

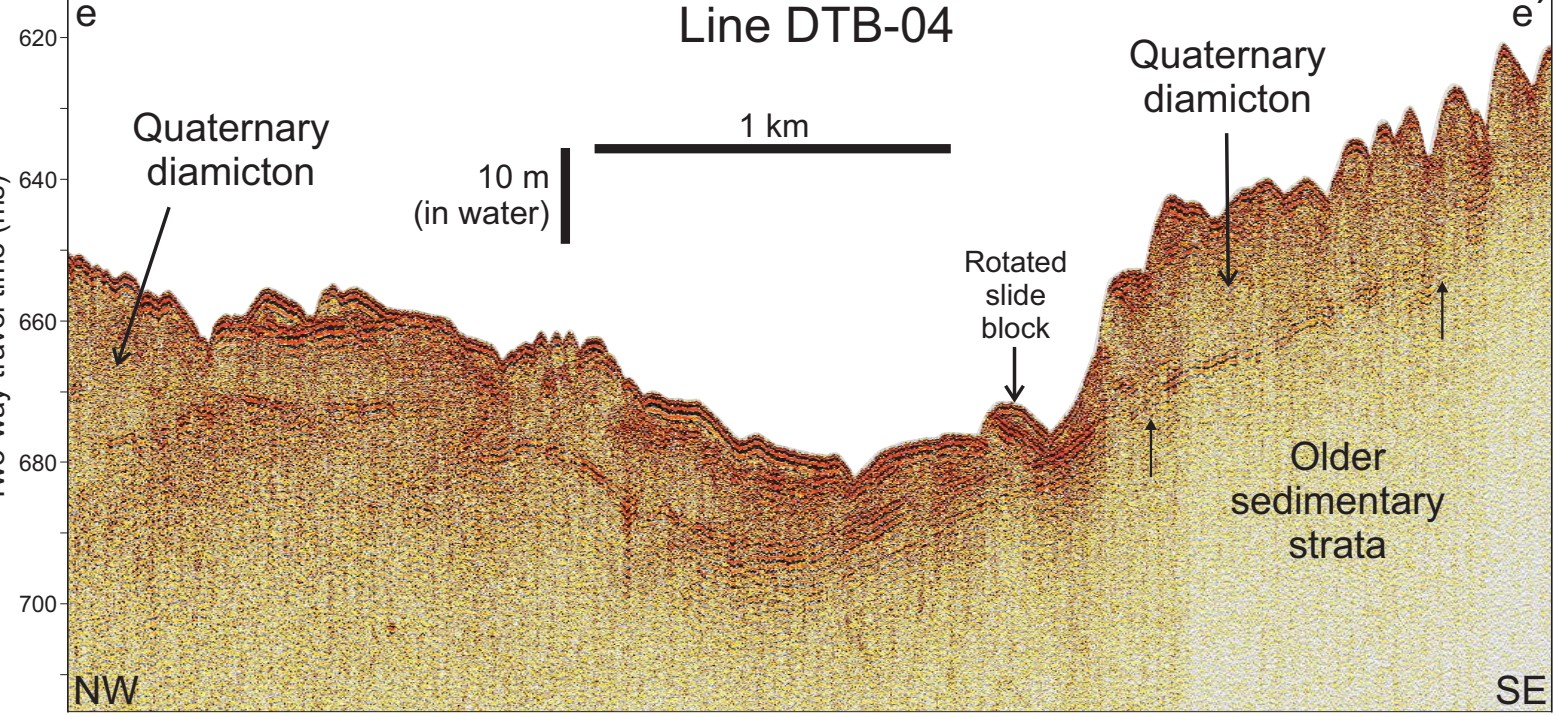

Fig. 6

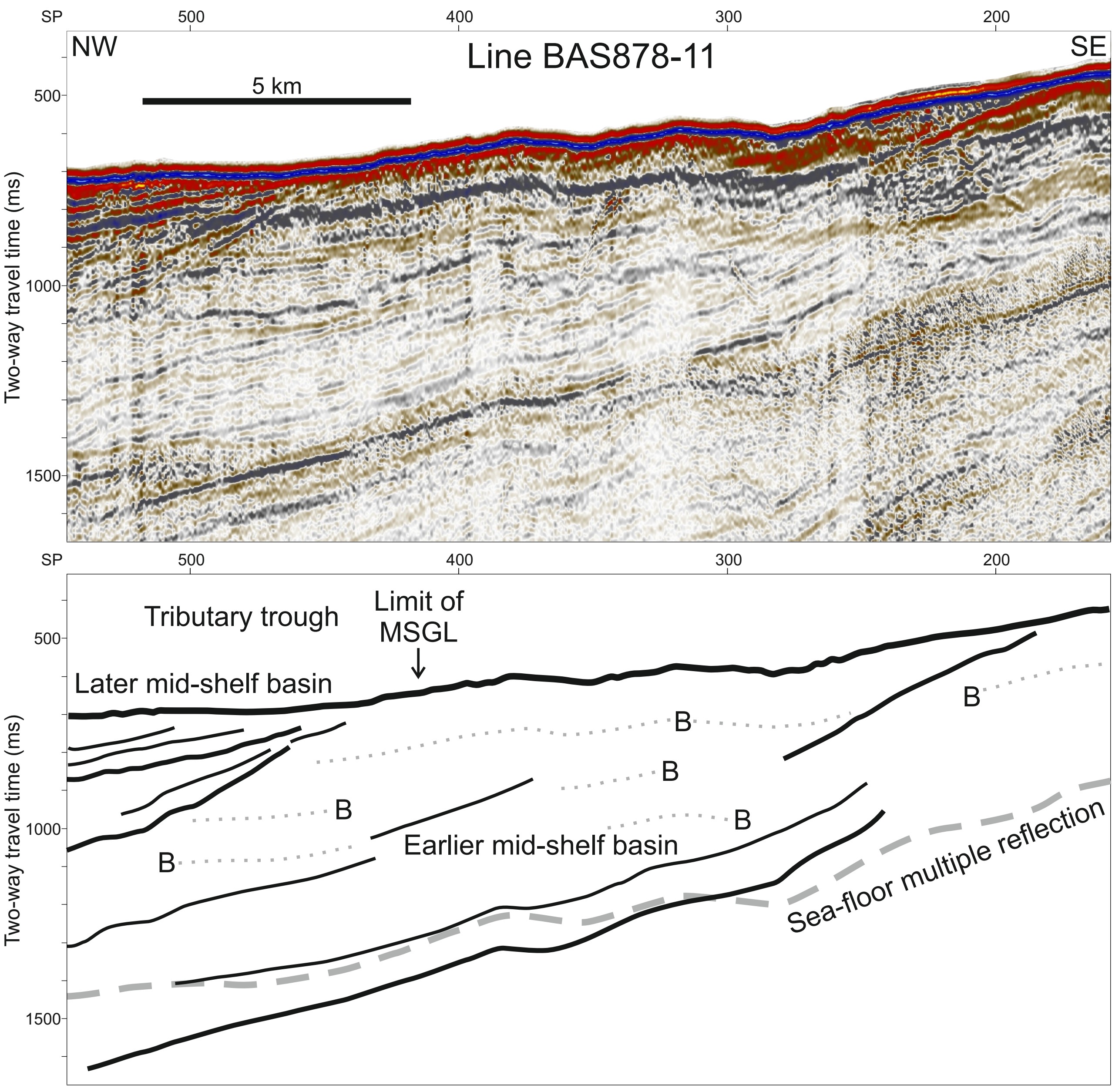

SP NW Line BAS878-11 SE

500 400 300 200

5 km

Two-way travel time (ms)

500

1000

1500

SP

500 400 300 200

Tributary trough    Limit of
MSGL

Later mid-shelf basin

Two-way travel time (ms)

500

B

B

B

B

B

B

Earlier mid-shelf basin

Sea-floor multiple reflection

1000

1500

Fig. 7