# Peer review of "1. Introduction"

_The Cryosphere, 2018_

## Referee Comment (RC1) · Anonymous Referee #1 · 6 Mar 2019

**Subglacial hydrological control on flow of an Antarctic Peninsula palaeo-ice stream. Larter et al.**

Larter et al present a combination of new and existing marine geophysical data and use these to interpret aspects of subglacial hydrology beneath the paleo ice sheet. This is an extremely useful application of data from the continental shelf due to the coverage and resolution of available data sets and the increased awareness of the role subglacial hydrology may play in ice-stream flow. I was especially pleased to see the inclusion of the high resolution acoustic sub-bottom (TOPAS) profile, and the legacy Deep Tow Boomer data. This study is well written and generally very well presented. Many of the comments I make in the following are intended to broaden the potential audience of the manuscript.

The introduction would benefit from an additional paragraph that briefly reviews the state of knowledge of subglacial land-forms and their significance for subglacial hydrology. I realize this is a large subject area but it's inclusion would widen the accessibility of the paper and provide more context for the observations and interpretation to come. Details on the ability of subglacial water to ascend an opposing bed slope, and modern ice sheet observations of subglacial water leading to ice sheet velocity changes [Siegfried and others, 2016, Stearns and others, 2008] could be included here. Also the implications of different distributions of water at the bed (concentrated versus distributed) could be highlighted.

**Minor Points**

There is an error in Figure 2 that led to some confusion. I think the seismic line location shown on Fig. 2 should be AMG 845-03 not BAS 878-11.

L25 either include the acronym (PD) here after Palmer Deep or omit the (AHT) on L21 (Then reconcile change with L58, L62).

L110–114 it would be worth stating the nominal resolution of these data here as the high resolution data offers a resolution in excess of what is presently possible beneath the ice sheets.

L126–127 this sort of content would be good in the suggested paragraph in the introduction.

L137 'where a MCS profile' to 'where a MCS profile (AMG 845-03)'

The Results section blends results and interpretation. Some sticklers would disagree with this approach as it doesn't allow the reader to digest the data on its own. But it is done well here

and I found it useful.

L196–197 'suggests that water supplied through them was important in lubricating and dilating...' Strictly this observation suggests that there was a change in the routing of water at the bed from an upstream regime dominated by flow though a major channel incised into the bed to a more distributed system with mobile subglacial till, that was likely dilatated, facilitating fast flow of the overriding ice. This observation would couple nicely with introductory/background material on the role of subglacial water in rapid ice flow.

Section 3.3 shows compelling evidence of shear margins beneath the palaeo ice-sheet. This evidence is more compelling than what we see beneath the modern ice sheet. Perhaps a statement along these lines in necessary to avoid future interpretations of shear margins being based on the high standards of these exemplar data.

L236–252 This paragraph is not as well linked to subglacial hydrology as earlier parts of the text. I appreciate that this is a unique look at an interesting subglacial landform but if the link to subglacial hydrology could be reinforced it would fit better. (L267 does this).

L273. It would be worth annotating the relevant profile with this location.

L288. Or surface slope reduction leading to subglacial flow change/reversal.

**Figures**
Fig 1. Make GZ dashed red lines thicker.
Fig 2. Change MCS label to AMG845-03.
Fig 4. Make these two the same scale.
Fig 6. Caption, 'small arrows' to upward pointing small arrows'
Fig 7. Mask the small annotations on the top axis in background of lower panel.

In closing, I thank the authors for their interesting study. I think the importance of bridging the gap between the marine and terrestrial ice sheet communities can not be over stated and I appreciate their efforts to do so.

Sincerely, Huw Horgan

**References**

[Siegfried and others, 2016] Siegfried, Matthew R., Helen A. Fricker, Sasha P. Carter and Slawek Tulaczyk, 2016. Episodic ice velocity fluctuations triggered by a subglacial flood in West Antarctica, *Geophysical Research Letters*, 2016GL067758.

[Stearns and others, 2008] Stearns, L. A., B. E. Smith and G. S. Hamilton, 2008. Increased flow speed on a large East Antarctic outlet glacier caused by subglacial floods, *Nature Geoscience*, **1**, 827–831.

---

## Referee Comment (RC2) · Anonymous Referee #2 · 14 Mar 2019

'Subglacial hydrological control on flow of an Antarctic Peninsula palaeo-ice stream' by Larter et al.

13 March 2019

**Review summary**

Larter et al. present new and legacy geophysical data from Anvers-Hugo Trough in the western Antarctic Peninsula. The bedforms indicate a strong control on ice flow and ice stream width (i.e., location of shear margins) due to the presence of basal meltwater. The study provides convincing evidence that channels drained water from a paleo subglacial lake residing in Palmer Deep, and downstream the channelized drainage fed a deforming bed with abundance mega-scale glacial lineations. The contrast between bedforms in Anvers-Hugo Trough and Perrier Troughs is striking – and an important set of observations of proximal ice streams with different basal thermal and hydrological regimes. I only have minor comments (listed below) and am in support of acceptance of the manuscript with minor revisions.

**Detailed comments**

Line 44: Spell out Anvers-Hugo Trough, as this is the first use in the main text.

Line 46: Identification of the lateral extent and connection with subglacial hydrology should be included in the abstract and highlighted more in the discussion.

Line 110: Quantify 'very high' resolution

Line 125: The focus is on subglacial hydrology; however, a brief morphological description of msgls and drumlins is needed, such as elongation ratios or lengths.

Line 182: What is the potential volume capacity of the Palmer Deep subglacial lake? It would be nice to insert here a comparison with identified active subglacial lakes and their timescale of drainage.

Line 196: Add further discussion on the nature of the transition between valleys (e.g., channels) and the onset of msgls, which is important for understanding the spatial transition of water drainage style (channelized versus darcian).

Line 307: Sedimentological evidence to support underlying strata as source of till?

Lines 314-320: This interpretation (or better suggestion) needs a more detailed description on the style of basal meltwater drainage and its influence on ice flow as documented in other studies. With that said, I do not think there is any evidence presented in this study to suggest that surging or flow acceleration occurred in response to lake drainage events and find this bit 'arm-wavy', which is unlike the rest of the text that is well-documented and supported.

Line 337: I don't follow this sentence. Where does the decadal to centennial timescale come from and how is it relevant here?

---

## Author Comment (AC1) · 23 Apr 2019

We thank Huw Horgan and an anonymous referee for their helpful comments on the manuscript.

Below we list our responses to each of the comments and details of the changes made to the manuscript. Reviewers' comments are listed in sequence followed in each case by our responses.

Comments by Huw Horgan

The introduction would benefit from an additional paragraph that briefly reviews the state of knowledge of subglacial land-forms and their significance for subglacial hydrology. Details on the ability of subglacial water to ascend an opposing bed slope,

and modern ice sheet observations of subglacial water leading to ice sheet velocity changes [Siegfried and others, 2016, Stearns and others, 2008] could be included here. Also the implications of different distributions of water at the bed (concentrated versus distributed) could be highlighted.

Two new paragraphs with some additional references addressing the points raised above have been inserted: "Knowledge and interpretations of submarine glacial landforms have advanced rapidly over the past two decades (Dowdeswell et al., 2016). For example, it is now recognised that elongated drumlins and mega-scale glacial lineations (MSGL) are signatures of past streaming ice flow on wet-based, mainly sedimentary beds, and that elongation generally increases with increasingly fast past flow rates (Stokes and Clark, 1999; Ó Cofaigh et al., 2002). The degree of preservation of fields of MSGL and the extent to which they are overprinted by grounding zone wedges and transverse moraines provides an indication to the rapidity of past grounding line retreat (Ó Cofaigh et al., 2008). 'Hill-hole pairs' and sediment rafts are recognised as features formed by glaciotectonic processes beneath cold, dry based ice (Evans et al., 2006; Klages et al., 2015). Erosion of meandering or anastomosing seabed channels and tunnel valleys with reversals of gradient along their length requires a hydrological pressure gradient that indicates they could only have formed beneath grounded ice (Ó Cofaigh et al., 2002; Nitsche et al., 2013). Simple hydrological pressure calculations indicate that a gentle ice surface gradient produces a pressure gradient at the bed that will drive water up an opposing bed slope nearly ten times as steep. "Modern ice sheet observations have revealed increases in ice flow rates over timescales of days to years in response to Antarctic subglacial drainage events (Stearns et al., 2008; Siegfried et al., 2016). Responses of glaciers in southwest Greenland to seasonal drainage of supraglacial meltwater to the bed, however, show that the mode of subglacial drainage is important, as a slow-down of glacier flow above a certain run-off threshold has been interpreted to correspond to a switch to more efficient, channelized drainage (Sundal et al., 2011)."

There is an error in Figure 2 that led to some confusion. I think the seismic line location shown on Fig. 2 should be AMG 845-03 not BAS 878-11.

The line number annotation on the figure has been corrected.

L25 either include the acronym (PD) here after Palmer Deep or omit the (AHT) on L21 (Then reconcile change with L58, L62).

The abbreviation AHT has been deleted from the abstract and both abbreviations are now introduced in the Introduction.

L110-114 it would be worth stating the nominal resolution of these data here as the high resolution data offers a resolution in excess of what is presently possible beneath the ice sheets.

The following sentence has been inserted: "The DTB system is capable of resolving sedimentary layers <1 m in thickness and also achieves very high spatial resolution dues to the proximity of the source and receiver to the sea bed."

L126-127 this sort of content would be good in the suggested paragraph in the introduction.

The comment was helpful in thinking about the paragraph added to the introduction, but the specific content here describes features shown by the data and therefore needs to be retained where it is.

L137 'where a MCS profile' to 'where a MCS profile (AMG 845-03)'

This text has been changed as suggested.

The Results section blends results and interpretation. Some sticklers would disagree with this approach as it doesn't allow the reader to digest the data on its own. But it is done well here and I found it useful.

We appreciate this comment. In initial drafts the description of landforms and interpre-

tation were written as separate sections but we decided that integrating them would streamline the text and make it easier to follow.

L196-197 'suggests that water supplied through them was important in lubricating and dilating...' Strictly this observation suggests that there was a change in the routing of water at the bed from an upstream regime dominated by flow though a major channel incised into the bed to a more distributed system with mobile subglacial till, that was likely dilatated, facilitating fast flow of the overriding ice. This observation would couple nicely with introductory/background material on the role of subglacial water in rapid ice flow.

We have inserted the following sentence to state this inference explicitly, and have also mentioned concepts of how channelized to distributed water flow at an ice bed affect ice flow in the introduction: "Thus we infer that the northward terminations of the valleys were associated with a transition from channelized to distributed water flow at the ice bed."

Section 3.3 shows compelling evidence of shear margins beneath the palaeo ice-sheet. This evidence is more compelling than what we see beneath the modern ice sheet. Perhaps a statement along these lines in necessary to avoid future interpretations of shear margins being based on the high standards of these exemplar data.

This comment is gratifying as one of the objectives we set out when we proposed this work was to find good examples of palaeo-shear margins. We have inserted the following sentence and associated reference: "Such clear expressions of ice stream shear margin positions in bed morphology have proved elusive beneath the modern ice sheet, partly because the chaotic structure of shear zones causes scattering of radar signals that is observed as 'clutter' in survey data (Shabtaie and Bentley, 1987)."

L236–252 This paragraph is not as well linked to subglacial hydrology as earlier parts of the text. I appreciate that this is a unique look at an interesting subglacial landform but if the link to subglacial hydrology could be reinforced it would fit better. (L267 does

this).

The following sentence has been inserted to clarify the inferred link between the observations reported here and subglacial hydrology: "As mentioned above, such holes are generally regarded as characteristic of erosion beneath dry-based ice cover, and therefore the restriction of erosion to the Quaternary diamictons is consistent with the shear stress threshold for brittle failure in them likely being significantly lower than in the underlying consolidated strata (Evans et al., 2006)." Additionally, the words "erosion and deformation beneath" have been inserted in a related sentence in the previous paragraph to clarify the processes that are thought to be involved.

L273. It would be worth annotating the relevant profile with this location.

The 'limit of MSGL' and the strata of different ages are already marked on Fig. 7. A call-out to the figure has been inserted at the end of the sentence.

L288. Or surface slope reduction leading to subglacial flow change/reversal.

This suggested alternative explanation has been incorporated into the sentence.

Fig 1. Make GZ dashed red lines thicker.

This has been done.

Fig 2. Change MCS label to AMG845-03.

This has been done.

Fig 4. Make these two the same scale.

The lower profile has been compressed horizontally to make it the same scale as the upper one.

Fig 6. Caption, 'small arrows' to upward pointing small arrows'

This has been done.

Fig 7. Mask the small annotations on the top axis in background of lower panel.

This has been done.

Comments by anonymous reviewer

Line 44: Spell out Anvers-Hugo Trough, as this is the first use in the main text.

This has been done.

Line 46: Identification of the lateral extent and connection with subglacial hydrology should be included in the abstract and highlighted more in the discussion.

The following sentence has been inserted in the abstract: "The data reveal a diverse range of landforms, including streamlined features where there was fast flow in the palaeo-ice stream, channels eroded by flow of subglacial water, and compelling evidence of palaeo-ice stream shear margin locations." The existing sentence that follows makes the connection between lateral extent and subglacial hydrology. The identification of the lateral extent is highlighted more in the discussion by addition of a sentence, as mentioned above in a response to a comment from the other reviewer, that describes how such clear expressions of ice stream shear margin positions in bed morphology have proved elusive beneath the modern ice sheet.

Line 110: Quantify 'very high' resolution

This has been done through addition of a sentence, as mentioned above in the response to the other reviewer, which states the capability of the DTB system to resolve sedimentary layers less than 1 m in thickness.

Line 125: The focus is on subglacial hydrology; however, a brief morphological description of msgls and drumlins is needed, such as elongation ratios or lengths.

The following sentences have been inserted: "Fields of drumlins, with elongation ratios between 2.5 and 6:1, occur over a transition zone between the rugged inner shelf and smoother mid-shelf part of the AHT, and where the AHT crosses a structural high that

Interactive
comment

separates middle and outer shelf areas (Larter et al., 1997). MSGL in the mid-shelf part of the AHT have elongation ratios between 12 and 17:1, whereas some on the outer shelf have elongation ratios up to 80:1."

Line 182: What is the potential volume capacity of the Palmer Deep subglacial lake? It would be nice to insert here a comparison with identified active subglacial lakes and their timescale of drainage.

The following paragraph has been inserted to address this comment: "A conservative estimate for the volume of the PD subglacial lake stated by Domack et al. (2006) was 20 km3. However, the full volume of the basin deeper than the sill depth of ~500 m is about 110 km3. If this was filled the lake would have been nearly two orders of magnitude greater in volume than subglacial Lake Ellsworth (1.37 km3, Woodward et al., 2010), but still nearly two orders of magnitude smaller than Lake Vostock (5400 ±1600 km3, Studinger et al., 2004). The length and width of PD are similar to Lake Engelhardt, the largest of a number of connected shallow lakes beneath the lower Mercer and Whillans ice streams, from which remote sensing data show ~2 km3 of water drained between October 2003 and June 2006 (Fricker et al., 2007). However, water depth in these lakes likely rarely exceeds 10 metres (Christianson et al., 2012), so their volumes are small compared to the potential size of lakes in deep basins such as PD."

Line 196: Add further discussion on the nature of the transition between valleys (e.g., channels) and the onset of msgls, which is important for understanding the spatial transition of water drainage style (channelized versus darcian).

The comment has been addressed through the addition of a sentence, as mentioned above in a response to a comment from the other reviewer, which states that we infer that the northward terminations of the valleys were associated with a transition from channelized to distributed water flow at the ice bed. The valley terminations are depicted clearly in the map and cross sections in Fig. 2, so we do not think there would
be any value in inserting additional descriptive text.

Line 307: Sedimentological evidence to support underlying strata as source of till?

Unfortunately, the underlying strata have never been sampled, so it is not possible to answer this question definitively through provenance analysis of till samples. However, reworked Cretaceous radiolarians were recorded in diamictons recovered by drilling on on the outer shelf on Ocean Drilling Program and these strata are the most liklely source. A sentence has been inserted mentioning this: "Although these strata have never been sampled they are the most likely source of reworked Cretaceous radiolarians found in diamictons recovered by drilling on the outer shelf (Shipboard Scientific Party, 1999)."

Lines 314-320: This interpretation (or better suggestion) needs a more detailed description on the style of basal meltwater drainage and its influence on ice flow as documented in other studies. With that said, I do not think there is any evidence presented in this study to suggest that surging or flow acceleration occurred in response to lake drainage events and find this bit 'arm-wavy', which is unlike the rest of the text that is well-documented and supported.

We acknowledge that this is speculation, but we contend that if there was episodic water discharge into the bed of the ice stream observations from modern glaciers indicate that this would likely have affected flow rates. The following sentence has been inserted to reinforce this point: "Acceleration of ice flow by about 10% over 14 months was observed on Byrd Glacier following a drainage event from subglacial lakes 200 km upstream of the grounding line (Stearns et al., 2008), and subglacial meltwater drainage flux and routing is also known to influence flow rates of glaciers in Greenland (Sundal et al., 2011)." If this inference is correct then, on that basis of the well-established Marine Ice Sheet Instability hypothesis it logically follows that variations in flow rate are likely to be associated with fluctuations in grounding line positions.

Line 337: I don't follow this sentence. Where does the decadal to centennial timescale

come from and how is it relevant here?

The timescale is based on the conceptual, semi-quantitative model in Alley et al. (2006). In fact that paper suggests the repeat period for outburst floods 'should be of the order of a few centuries' for the case of a hypothetical subglacial lake with similar dimensions to Palmer Deep. The first part of the sentence has been changed to make this clearer and to cite the Alley et al. paper, and now read as follows: "Such behaviour of marine-based palaeo-ice streams on timescales of the order of centuries, as suggested by the simple model proposed by Alley et al. (2006), . . ."

Figures to which minor chnages have been made following suggestions made by the referees are included below.

A revised version of the manuscript with edits made in MS Word Track Changes is attached as a supplementary zip file.

Please also note the supplement to this comment:
https://www.the-cryosphere-discuss.net/tc-2018-273/tc-2018-273-AC1-supplement.zip
* * *
[Figure]

**Fig. 1.**

[Figure]

Fig. 2

**Fig. 2.**

[Figure]

[Figure]

Fig. 4

**Fig. 3.**

[Figure]

Fig. 7

**Fig. 4.**